# Assessment of Investment Attractiveness of Projects on the Basis of Environmental Factors

**Sayabek Ziyadin** [1,*], **Elena Streltsova** [2], **Alex Borodin** [3], **Nataliya Kiseleva** [4], **Irina Yakovenko** [2] and **Elmira Baimukhanbetova** [1]

[1]   Al-Farabi Kazakh National University, The Center for Economic Research, Almaty 050040, Kazakhstan; ertis_economika@mail.ru
[2]   Department of Computer Software, Platov South Russian State Polytechnic University (NPI), 346428 Novocherkassk, Russia; el_strel@mail.ru (E.S.); iranyak@mail.ru (I.Y.)
[3]   Department of Finance, Plekhanov Russian University of Economics, 101000 Moscow, Russia; aib-2004@yandex.ru
[4]   Department of Finance, Russian Academy of National Economy and Public Administration at the PRESIDENT of the Russia Federation, 119571 Moscow, Russia; kiseleva-n-n@yandex.ru
[*]   Correspondence: sayabekz@gmail.com

**Abstract:** This article is devoted to the creation of intelligent modelling tools for decision support in the evaluation of intellectual projects submitted for financing, as based on qualitatively defined characteristics. The economic and mathematical models that form the basis of the toolkit are constructed using the mathematical apparatus of fuzzy logic, which allows for the description of poorly structured knowledge of specialists, as well as their application in solving questions about the extent of the impact of the proposed projects on the environment. The authors classify investment projects according to the degree of impact on the environment, the environmental criteria required by the investor for the evaluation of investment projects, and the formal formulation of the problem of evaluation of investment projects when taking into account the environmental factor. The toolkit was created based on the concept of intellectualization, where economic and mathematical models for the evaluation of investment projects are programmatically implemented via the tools and functions available in the MATLAB package.

**Keywords:** investment project; semi-structured problem; mathematical model; fuzzy logic; decision support; sustainable development

---

## 1. Introduction

The central issue for the economy of any country is that of increasing its rate of economic growth, a reliable driver of which is the formation and development of a strategy for the sustainable development of territories based on the intensification of investment activities. Sustainable development refers to the process of economic and social changes in which a balance is achieved between the exploitation of natural resources, investment activities, and scientific and technological development.

In recent years, considerable research has been conducted into the development of the digital economy. Thus, the Fletcher School at Tufts University, in partnership with Mastercard, presented the research "The Digital Evolution Index 2017" [1]. The companies analysed the progress that countries have made in the development of their digital economies, as well as the degree of integration of new opportunities into people's daily lives. According to the research, Singapore, the United Kingdom, New Zealand, the United Arab Emirates, Estonia, Hong Kong, Japan and Israel are leaders in the development of the digital economy. The pace of digital evolution in these countries serves as an example for other countries in their choice of development vector.

According to the results of the Mastercard survey, which was conducted among online users living in 60 countries worldwide, an index of users of Internet technologies was calculated, which was determined on the basis of four key indicators and 170 unique indicators: access to the Internet and infrastructure development; consumer demand for digital technology; public policies, laws and resources in this area; and innovations in the country (including investments in technology and digital start-ups). These studies were conducted in Norway, Sweden, Switzerland, Denmark, Finland, Singapore, South Korea, the United Kingdom, Hong Kong and the United States, which are among the 10 countries worldwide with the most developed digital economies. After analysing the growth rate and state of the digital economy, countries are divided into four categories:

First category—leading countries: Singapore, UK, New Zealand, UAE, Estonia, Hong Kong, Japan and Israel demonstrate high rates of digital development and continue to lead in the spread of related innovation.

Second category—countries with slower growth rates: This includes many of the developed countries of Western Europe, the Scandinavian countries, as well as Australia and South Korea. For a long time, these countries have showed steady growth but at present they have significantly reduced their pace of development. Without innovation, they risk falling behind those leading digitalization. As can be seen, the two largest world economies (USA and Germany) are on the border of being leading countries and slowing countries, followed closely by the third-largest economy in the world, Japan. At the same time, the dynamics of digital development in the UK are greater than in all other countries of the European Union.

Third category—promising countries: In spite of the fact that these countries demonstrate a relatively low overall level of digitalization, they demonstrate steady growth rates that are attracting investors. These countries include China, Kenya, Russia, India, Malaysia, Philippines, Indonesia, Brazil, Colombia, Chile and Mexico.

Fourth category—problem countries: These include such counties as South Africa, Peru, Egypt, Greece, and Pakistan. These countries face serious challenges that are associated with low levels of digital development and slow growth. For the leadership of these countries, it is necessary to realize the risk of being left in a "digital deadlock" and to examine, as per the example of countries with more dynamic development, which policy measures could increase their countries' competitiveness [2].

A decisive role in achieving the balance between the techno sphere and the natural sphere, in preserving natural capital, along with the development of the techno sphere, is played by digital technologies. At the same time, the environmental aspect of the digital economy, as the most important component of sustainable development, is of particular importance. In the framework of the sustainable development of territories, a prerequisite for the effective development of their economic growth strategies is the intensification of innovation and investment activities in the formation and development of production potential. In the course of selecting investment projects, the creation of a digital platform, in the form of a set of mathematical models and software based on these models, makes it possible to reduce the risks associated with such projects while taking environmental factors into account, and which is otherwise an urgent problem.

The development of any country is determined by solving the tasks associated with the formation of effective territorial strategies designed to accelerate economic growth, a necessary condition which requires extensive investment activity. When forming the production potential of territories on a new scientific and technical base through the process of innovation, investment projects predetermine the competitiveness of a country's regions and, consequently, the country as a whole. Along with solving global problems related to economic and social progress, one of the most important aspects of the innovation and investment concept of regional development is to minimize the severity of environmental problems and to focus on the international doctrine of sustainable development. In this regard, the consideration of environmental factors has become integral to the evaluation of investment projects.

The growing importance and relevance of environmental factors as part of the conditions of digital transformation has led to the choice of the current research topic.

## 2. Literature Review

Analysis of publications on the problem of sustainable development of territories and organizations, as well as the tools used in this aspect, has revealed a wide range of studies, including the work of Alekseenko [3], Tychinina [4], Ryabov [5], and Korobkova [6]. The authors of these studies have focussed on a systematic approach to monitoring the sustainable development of organizations as the main aspect of the territorial economy. The research of Ziyadin and Kabasheva [7] has considered the methodology required to identify the most important factors of sustainable development to reduce the negative impact of some of them and to determine the effectiveness of policies that contribute to the sustainable development of industrial and economic activity. The system of aggregated indicators of the sustainable development of enterprises and the formation of operation areas under modern conditions have been proposed by Rodionova & Abdulina [8] and Kucherova [9]. Scientific and methodological approaches to the sustainable development of organizations, as well as the systematization of factors of sustainable development, have been described by Anpilov [10] and Bat'kovskiy [11]. The prospects for the evolution of the state using models of sustainable development have been considered by Ursul [12], Ivanov [13], TSapieva [14], Rozenberg, CHernikova, Krasnoschekov, Krylov & Gelashvili [15], and Lyubushin, Babicheva, Galushkina & Kozlova [16]. The experience of the strategic management of sustainable development with structural reforms in East Asian countries was revised in the studies by Ziyadin, Kabasheva, Suieubayeva, and Moldazhanov [17]. Ziyadin, Khamitova, Khassenova, Suieubayeva, and Agumbayeva [18] examined the policy of diversification of industry and services in Asia as a tool for sustainable development. SHalmuev [19], Ugol'nitskiy [20], SHevchenko & Litvinskiy [21], and Ziyadin, Omarova, Doszhan, Saparova & Zharaskyzy [22] studied issues of sustainable development in conjunction with the problems of management and diversification of the results of R&D. Mathematical models have been actively used during the selection of appropriate development schemes. In the process of the digitalization of the economy, the problems of applying mathematical modelling methods to solving problems of sustainable development are becoming increasingly important. Mathematical modelling of the world economy in terms of sustainable development has been given considerable attention by researchers such as Makhov [23]. The directions which the sustainable development of territories based on innovation have taken, as well as the application of intellectual decision support methods, are presented in the works of Zakharova [24] and Kolosova & KHavin [25]. According to Badulescu, Bungau & Badulescu [26], the task of introducing a sustainable development model is effectively that of promoting it as the main driving force for sustainability-oriented enterprises, that is, firms that meet profitability, environmental and social requirements. In more recent research, the theoretical and practical aspects of informatization and digitalization, alongside socio-economic and environmental problems in different sectors of the economy, are widely presented. For example, Shaikh & Karjaluoto [27] use the broad term "information technology/systems" to refer to a set of systems, technologies, processes, business applications, and software. In this regard, the work of Watkins, Ziyadin, Imatayeva, Kurmangalieva & Blembayeva [28], Ursul [29], Kanin, Parinova & L'vovich [30], Averchenkov, A.A.; Maksimenko, Y.L. [31], Borodin, A., Shash, N., Kiseleva N. [32], Ilina, I., Streltsova, E., Yakovenko, I. [33] should be noted. Despite the importance of the approaches, methods, models, and technologies developed to support decision making in the field of sustainable development, it is clear that when taking into account environmental factors, the problems of mathematical modelling in this area have been insufficiently researched. This is especially true of the problems of intellectualization of the model tools that function in conditions of uncertainty and allow the processing of poorly structured knowledge of natural intelligence.

## 3. Methodological Framework

### 3.1. Problem Statement

Implementing an approach to changing the environmental strategy from one of environmental concerns to preventive, the authors of the article set the task of creating a toolkit for assessing the environmental attractiveness of investment projects, carrying out a two-stage procedure that allows for environmental screening in the form of an economic-mathematical model, *MOD*, to determine the compliance of investment projects with the environmental criteria as well as allowing the financing of investment projects selected during the implementation of the first stage to be prioritized. The proposed toolkit for assessing environmental attractiveness (acceptability) at the environmental screening stage is a complex of economic and mathematical models:

$$MOD = \langle M_1, M_2 \rangle \tag{1}$$

where $M_1$ is the model for determining the compliance of investment projects with environmental criteria, and $M_2$ is the model used to identify the priority of an environmental project.

According to the classification given by Averchenkov & Maksimenko [31], each investment project is categorized as A, B, C, or D, which identifies the degree of its impact on the environment (Table 1).

**Table 1.** Classification of investment projects by environmental impact.

| Categories of Investment Projects | Designation from the Model | Criteria for Categories of Investment Projects |
|---|---|---|
| A | *A* | Implementation of the investment project may lead to irreversible environmental consequences. |
| B | *B* | The implementation of an investment project may lead to adverse environmental impacts (one or more natural components), but these impacts are easily recognizable and can be avoided by applying environmental or countervailing measures. |
| C | *W* | The implementation of the investment project will not have an adverse impact on the environment and will not lead to any adverse effects. |
| D | *G* | The investment project is not related to industrial production and involves the improvement of the environment. |

In addition to the assessment by category, the investment projects submitted for financing are evaluated according to the individual criteria of the investor. Examples of such criteria are given in Table 2.

The priority of investment projects is determined by the number of points awarded for each of the following four criteria:

(1) the scale of the impact on the environment, both existing and overcome through the implementation of the investment project;
(2) objects of adverse effects that the implementation of the investment project is aimed at overcoming;
(3) the environmental situation in the territory of the investment project;
(4) the type of reduced (preventable) environmental impact (Table 3).

**Table 2.** Examples of environmental criteria set by investors for the evaluation of investment projects.

| Criteria Designation | Characteristics of Criteria |
|---|---|
| *LIW* | The implementation of the investment project should lead to the elimination of the sources of environmental impact. |
| *REP* | The implementation of the investment project should be aimed at solving one of several environmental problems:<br>(a) air, groundwater, surface water and soil pollution;<br>(b) accumulation of hazardous and other wastes;<br>(c) depletion or destruction of natural resources;(d) changes in traditional land use;<br>(d) changes in natural landscapes, etc. |
| *ND* | The implementation of the investment project should not lead to an adverse impact on the environment due to:<br>(a) increasing output;<br>(b) fundamental changes in the underlying technology;<br>(c) the need to increase the volume of raw materials used, which are minerals;<br>(d) use of non-renewable natural resources. |

**Table 3.** Criteria for determining the priority of the investment project.

| Criteria for Determining the Priority of the Investment Project | | | Score |
|---|---|---|---|
| SCALE OF ENVIRONMENTAL IMPACT (*SEI*) | 1.1 | National: covers the economic regions or territory of several regions. | 7 |
| | 1.2 | Regional: large city, region. | 5 |
| | 1.3 | Provincial: district, village, rural district. | 3 |
| | 1.4 | Local: industrial zone of the enterprise. | 2 |
| IMPACT OBJECT (*IO*) | 2.1 | Public safety: long-term pollution of the environment, causing statistically recorded indicators of deterioration in the health of the population and threat to livelihoods. | 9 |
| | 2.2 | Public health: environmental pollution, which may lead to a deterioration in the health of the population. | 6 |
| | 2.3 | Individual natural components: water bodies, atmospheric air, soils, forests, etc. | 5 |
| | 2.4 | Natural resources: minerals, underground and surface waters, flora and fauna. | 3 |
| ENVIRONMENTAL SITUATION IN THE PROJECT AREA (*ES*) | 3.1 | Extremely unfavourable: according to long-term observations, the state of the environment is assessed by environmental authorities as extreme. | 9 |
| | 3.2 | Unfavourable: indicators of the state of the environment or its individual components many times exceed the maximum permissible values. | 5 |
| | 3.3 | Generally favourable, but there are separate sources of pollution. | 2 |
| TYPE OF PREVENTABLE ENVIRONMENTAL IMPACT (*EI*) | 4.1 | Air pollution. | 9 |
| | 4.2 | Surface water pollution, groundwater pollution, pollution by hazardous industrial waste. | 6 |
| | 4.3 | Soil pollution. | 3 |
| | 4.4 | Noise, vibration, odours. | 1 |

It may be that an investment project does not meet the environmental criteria of the investor and is therefore not eligible for financing. Therefore, at the stage of selection of investment projects it would be necessary to use their expertise in order to avoid financial costs to projects that could cause harm to the environment. As the reader may be aware, the environmental information contained in the project funding application is very limited, and the specialist must establish the compliance of the project with environmental criteria based on their knowledge. Thus, expert assessments of investment projects should represent some symbiosis of the qualitative characteristics expressed in Tables 1–3, which, ultimately, should be transformed into quantitative indicators in order to rank projects according to their impact on the environment.

The problem is formulated in the following manner. As mentioned earlier, in order to prioritize investment projects, which at the previous stage were assigned by the $M_1$ model to classes A, B, C or D, according to the $M_2$ model, the task of determining the priority is set as follows: denote the set of investment projects classified as A, B, C, or D by $PR = \{PR_1, PR_2, \ldots, PR_n\}$. The system of input variables, on the basis of which the expert assessment of the investment priority of $PR$ projects is carried out, is denoted $\{SEI, IO, ES, EI\}$, where $SEI$ is the scale of the environmental impact, $IO$ are objects of adverse impact due to the implementation of the investment project, $ES$ are indicators of the environmental situation in the investment project territory, and $EI$ are characteristics of the type of reduced (preventable) environmental impact. The objective of the investment project evaluation is to determine, for any combination of values of input variables $\{SEI, IO, ES, EI\}$ set by experts, the level of priority of the $URS$ investment project (Figure 1).

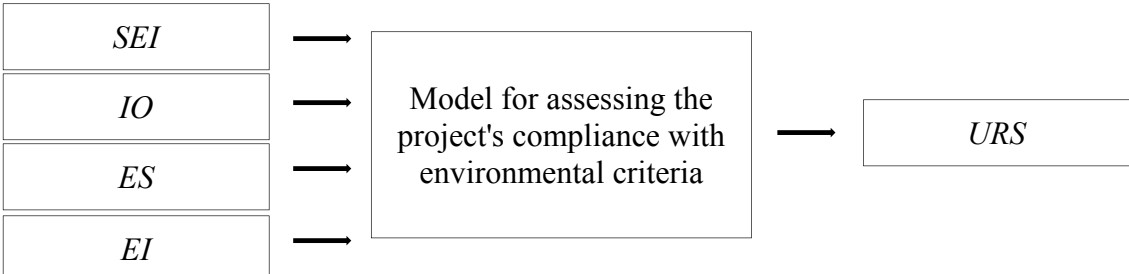

**Figure 1.** Setting the task of assessing the degree of compliance of the project submitted for financing with environmental criteria.

In Figure 1, the output variable $URS$ is an integral indicator of the degree of compliance of the investment project submitted for financing with environmental criteria.

As can be seen from Tables 1 and 2, the information characterizing the project is poorly formalized. Therefore, the tools used for project evaluation should operate with qualitatively defined indicators of sets $\{SEI, IO, ES, EI\}$, transforming them into an integral indicator $URS$. The tools created for the evaluation of investment projects should be based on the concept of intellectualization, which consists of providing the opportunity to formalize expert knowledge, and their processing and application in the decision-making process. In the modern scientific literature, there are known approaches to the creation of intelligent decision support tools. In Tychinina [4], Ryabov [5], and Korobkova [6], intelligent components of intelligent models with learning properties are realized based on the application of the theory of stochastic automata functioning in random media.

As an example of the development of intellectual tools, we can cite the models of strategic management of sustainable development for industrial enterprises [8]. However, the proposed approaches, models, and algorithms do not solve all the problems of intellectualization and do not entirely negate the possibility of their addition with new proposals.

## 3.2. Problem-Solving Method

The issues of building an economic and mathematical toolkit to evaluate investment projects in line with environmental safety, as based on the use of the mathematical apparatus of fuzzy algebra, were developed in the work of various scientists. Therefore, Verekhin & YAchmeneva [34] proposed mathematical models through which to find quantitative estimates of the level of ecological and economic security of regional industrial enterprises, using the adaptive tools of fuzzy sets. Based on the application of a balanced system of indicators and characteristics of environmental and economic security, a system for assessing environmental safety has been built based on the formalization and further use of the knowledge of specialists. Ivantsova & Kuz'min [35] used fuzzy logic methods to model the ecological and economic safety of production processes in industrial enterprises. The mathematical apparatus of fuzzy logic has also been used in combination with classical research methods and simulation modelling. An attempt to use fuzzy-logical mathematical methods in the study of the risks arising in the economic activity of enterprises to prevent damage and minimize potential losses was made by Rogachev, Shevchenko & Kuz'min [36]. The solution to the problem of constructing a multi-criteria model for determining the best available technology with fuzzy source data was proposed by Ptuskin, Levner & ZHukova [37]. To solve the problem posed, the representation of the parameter values in the form of fuzzy numbers is justified. The construction of expert systems based on a hierarchical fuzzy inference was noted by YAstrebova [38]. The synthesis of fuzzy decision rules for medico-ecological applications based on data structure analysis was described by Korenevskiy, Filist & CHursin [39]. The authors proposed a method of combining the membership functions of fuzzy sets to solve problems related to differential diagnostics for various types of medical problems. Tindova [40] considered the tasks of using fuzzy inference when solving a particular class of assessment task for environmental monitoring. Analysis of environmental data based on the use of fuzzy logic methods was suggested by Alekseev, Telegina & YAnnikov [41]. Fuzzy-multiple formalization of the balanced scorecard for offshore oil and gas companies was proposed by Nedosekin, Abdulaeva & SHkatov [42]. The analysis of the research results showed that the use of the mathematical apparatus of fuzzy logic is currently a very relevant generalization of classical logic in the construction of environmental safety systems. The mathematical apparatus of fuzzy logic allows one to embed weakly structured control rules into the system, obtained on the basis of the formalization of the a priori knowledge of experts and described in a form close to natural language. In this regard, the authors used the mathematical apparatus of fuzzy algebra and fuzzy logic in the construction of an economic–mathematical model for the formal description of the weakly structured knowledge of an expert in the environmental assessment of investment projects.

The task of papers in the building of membership functions is posed as follows. Two parameters are given: input parameters, $\{SEI, IO, ES, EI\}$, and output parameters, $URS$, which are treated as linguistic variables. The parameters are described by the class of mathematical apparatus of linguistic variables, which are tuples in the following form:

$$\begin{aligned}
&\langle SEI, \ T(SEI), \ U_{SEI}, \mu_{SEI}\rangle, \\
&\langle IO, \ T(IO), \ U_{IO}, \mu_{IO}\rangle, \\
&\langle ES, \ T(ES), \ U_{ES}, \mu_{ES}\rangle, \\
&\langle EI, \ T(EI), \ U_{EI}, \mu_{EI}\rangle, \\
&\langle URS, \ T(URS), \ U_{URS}, \mu_{URS}\rangle
\end{aligned} \tag{2}$$

where, $T(SEI) = \left\{A^i_{SEI}\right\}^\alpha_{i=1}$, $T(IO) = \left\{A^i_{IO}\right\}^\alpha_{i=1}$, $T(ES) = \left\{A^i_{ES}\right\}^\alpha_{i=1}$, $T(EI) = \left\{A^i_{EI}\right\}^\alpha_{i=1}$, $T(URS) = \left\{A^i_{URS}\right\}^\alpha_{i=1}$ are sets of terms of linguistic variables; $U_{SEI}, U_{IO}, U_{ES}, U_{EI}, U_{URS}$ are universal sets on which the membership functions are set $\mu_{SEI} = \{\mu_{A^i_{SEI}}\}^\alpha_{i=1}$, $\mu_{IO} = \{\mu_{A^i_{IO}}\}^\alpha_{i=1}$, $\mu_{ES} = \{\mu_{A^i_{ES}}\}^\alpha_{i=1}$, $\mu_{EI} = \{\mu_{A^i_{EI}}\}^\alpha_{i=1}$, $\mu_{URS} = \{\mu_{A^i_{URS}}\}^\alpha_{i=1}$ of their terms, considered to be the fuzzy sets $A^i_{SEI} = \int_{U_{SEI}} \mu^i_{SEI}/u$, $A^i_{IO} = \int_{U_{IO}} \mu^i_{IO}/u$, $A^i_{ES} = \int_{U_{ES}} \mu^i_{ES}/u$, $A^i_{EI} = \int_{U_{EI}} \mu^i_{EI}/u$, $A^i_{URS} = \int_{U_{URS}} \mu^i_{URS}/u$.

The notation for the terms *URS* linguistic variables {*SEI*, *IO*, *ES*, *EI*} are given in Table 3. For the output linguistic variable *URS*, a set of terms has been proposed that characterize the priority level of investment projects $A_{URS}^1 = L$, $A_{URS}^2 = M$, $A_{URS}^3 = H$ submitted for funding, meaning the priority levels "low", "medium", and "high" and which are rated on a three-point scale $U_{URS}$ [0, 3]. The universal sets $U_{SEI}$, $U_{IO}$, $U_{ES}$, $U_{EI}$ correspond to the rating scales given in Table 3.

Using the ideas of the appropriate experts, as formed based on their knowledge and experience, we assume that the task of determining the compliance of the project $PR_i$ with environmental criteria can be described by the following proposals in natural language:

(a) if the scale of the investment project's impact on the environment is national (in our notation, this is *SEI* = 1.1), regional (*SEI* = 1.2), local (*SEI* = 1.3) or local (*SEI* = 1.4), the implementation of the investment project is aimed at addressing the safety of the population (*IO* = 2.1) and the environmental situation in the territory is extremely unfavorable (*ES* = 3.1), then the investment project submitted for financing is given a high level of priority ($A_{URS}^3 = H$);

(b) if the implementation of an investment project is aimed at reducing the pollution of groundwater, reducing pollution by solid waste (*EI* = 4.2) and the environmental situation in the area is extremely unfavorable (*ES* = 3.1) or unfavorable (*ES* = 3.2), then the investment project is considered to be high priority level ($A_{URS}^3 = H$);

(c) if the environmental situation in the territory is generally favourable (*ES* = 3.3) and the project implementation does not reduce the adverse environmental impact (*EI* = *not* (4.1) or *EI* = *not* (4.2) or *EI* = *not* (4.3) or *EI* = *not* (4.4)), then the investment project should be attributed a low level of priority ($A_{URS}^1 = L$);

(d) if the scale of the project's impact on the environment is local (*SEI* = 1.4), the situation in the territory is generally favourable (*ES* = 3.3), and project implementation prevents noise and vibration (*EI* = 4.4), then the investment project is of average priority ($A_{URS}^2 = M$).

## 4. Results and Discussion

Let us set the semantics of fuzzy sets $A_{SEI}^i \in T(SEI)$ in the form of membership functions $\mu_{SEI} = \{\mu_{A_{SEI}^i}\}_{i=1}^{\alpha}$ of a trapezoidal type, assuming that their universums are segments: $U_{SEI} = [0,7]$: $\mu_{SEI}^{1.1}(u,0,7,7)$, $\mu_{SEI}^{1.2}(u,0,5,7)$, $\mu_{SEI}^{1.3}(u,0,3,7)$, $\mu_{SEI}^{1.4}(u,0,2,7)$. The membership functions entered have the following form:

$$\mu_{SEI}^{1.1}(u,0,7,7) = \begin{cases} 0, u < 0; \\ \frac{u}{7}, 0 \le u \le 7; \\ 0, u > 7; \end{cases} \qquad \mu_{SEI}^{1.2}(u,0,5,7) = \begin{cases} 0, u < 0; \\ \frac{u}{5}, 0 \le u \le 5; \\ \frac{7-u}{2}, 5 \le u \le 7; \\ 0, u \ge 7; \end{cases}$$

$$\mu_{SEI}^{1.3}(u,0,3,7) = \begin{cases} 0, u < 0; \\ \frac{u}{3}, 0 \le u \le 3; \\ \frac{7-u}{4}, 3 \le u \le 7; \\ 0, u \ge 7; \end{cases} \qquad \mu_{SEI}^{1.4}(u,0,2,7) = \begin{cases} 0, u < 0; \\ \frac{u}{2}, 0 \le u \le 2; \\ \frac{7-u}{5}, 2 \le u \le 7; \\ 0, u \ge 7. \end{cases}$$

In the MATLAB system of the Fuzzy Logic Toolbox package, these functions take the form shown in Figure 2.

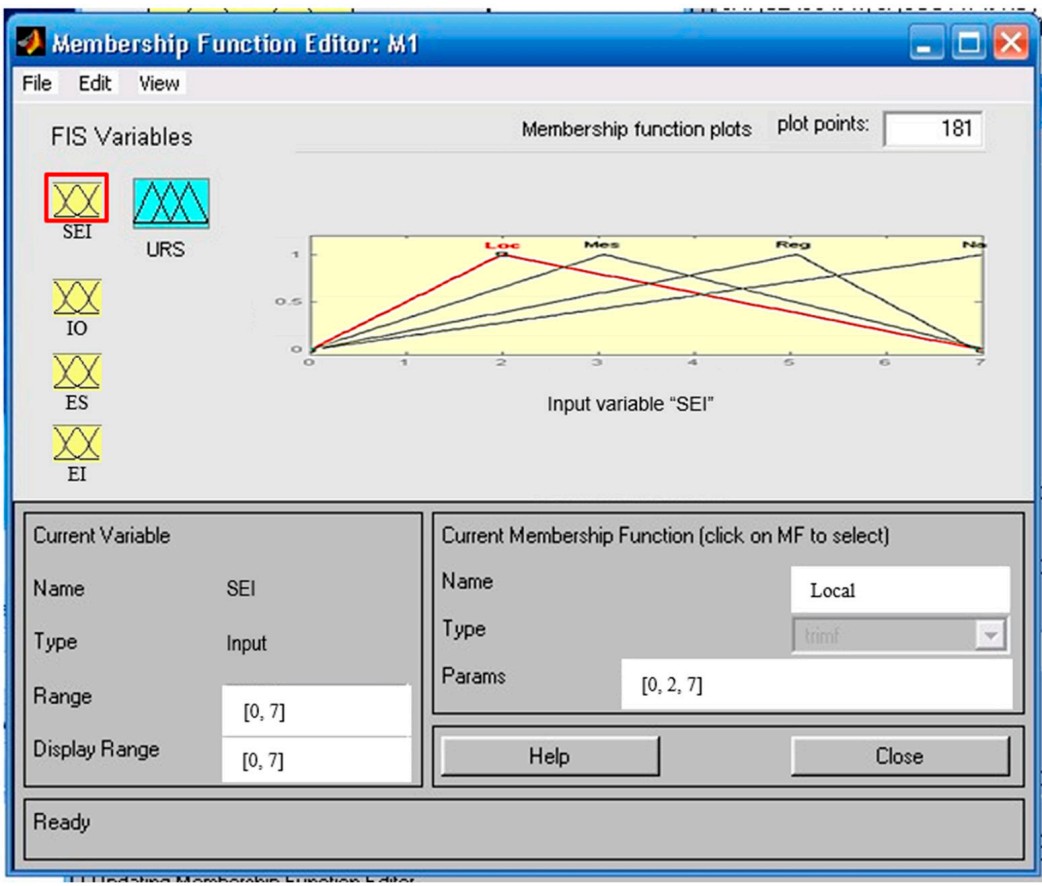

**Figure 2.** Graphs of membership functions of fuzzy sets of an example linguistic variable *SEI*.

Semantics of fuzzy sets $A_{IO}^i \in T(IO)$ in the form of membership functions $\mu_{IO} = \{\mu_{A_{IO}^i}\}_{i=1}^\alpha$ of the trapezoidal type, assuming that their universums are segments: $U_{IO} = [0, 9]$: $\mu_{IO}^{2.1}(u, 0, 9, 9)$, $\mu_{IO}^{2.2}(u, 0, 6, 9), \mu_{IO}^{2.3}(u, 0, 5, 9), \mu_{IO}^{2.4}(u, 0, 3, 9)$. The membership functions entered have the following form:

$$\mu_{IO}^{2.1}(u, 0, 9, 9) = \begin{cases} 0, u < 0; \\ \frac{u}{9}, 0 \le u \le 9; \\ 0, u > 9; \end{cases} \qquad \mu_{IO}^{2.2}(u, 0, 6, 9) = \begin{cases} 0, u < 0; \\ \frac{u}{6}, 0 \le u \le 6; \\ \frac{9-u}{3}, 6 \le u \le 9; \\ 0, u \ge 9; \end{cases}$$

$$\mu_{IO}^{2.3}(u, 0, 5, 9) = \begin{cases} 0, u < 0; \\ \frac{u}{5}, 0 \le u \le 5; \\ \frac{9-u}{4}, 5 \le u \le 9; \\ 0, u \ge 9; \end{cases} \qquad \mu_{IO}^{2.4}(u, 0, 3, 9) = \begin{cases} 0, u < 0; \\ \frac{u}{3}, 0 \le u \le 3; \\ \frac{9-u}{6}, 3 \le u \le 9; \\ 0, u \ge 9. \end{cases}$$

The graphs of these functions, shown in the MATLAB system environment of the Fuzzy Logic Toolbox package, have the form shown in Figure 3.

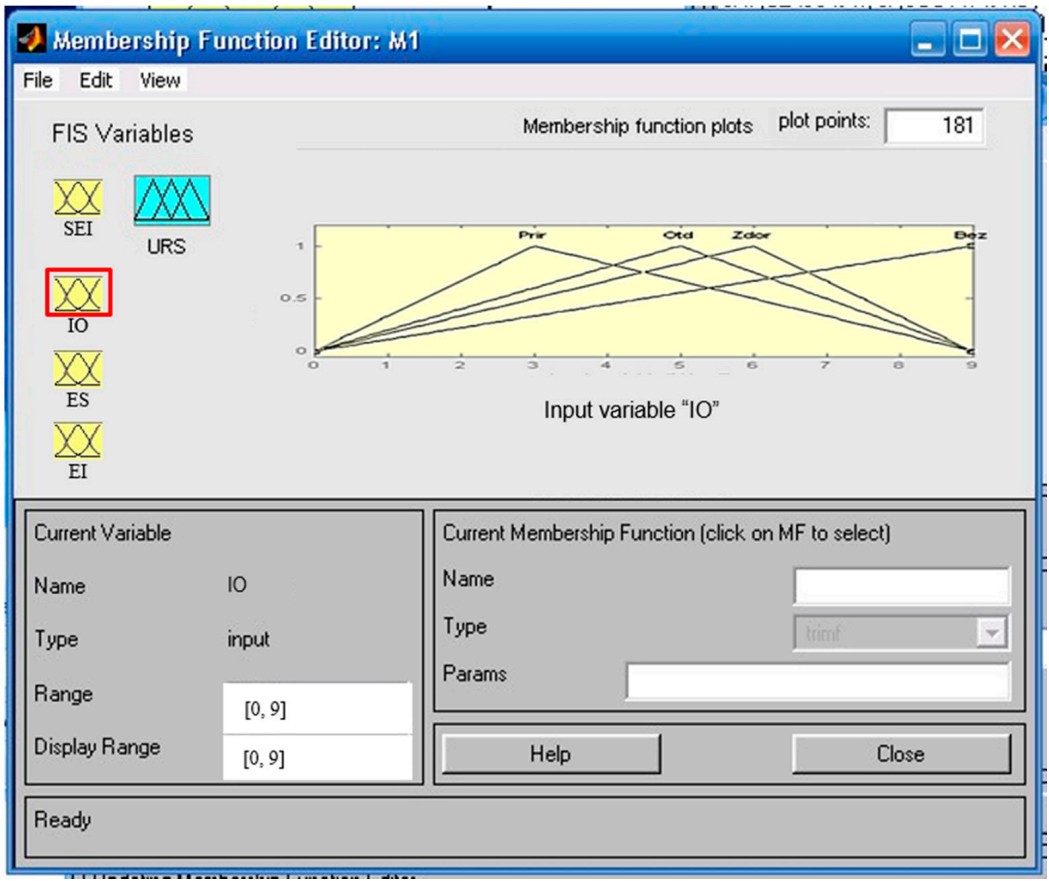

**Figure 3.** Graphs of membership functions of fuzzy sets of a linguistic variable *IO*.

The semantics of fuzzy sets of the linguistic variable *ES* are given in the form of triangular membership functions:

$$\mu_{ES}^{3.1}(u,0,9,9) = \begin{cases} 0, \, u < 0; \\ \frac{u}{9}, 0 \le u \le 9; \\ 0, u > 9; \end{cases} \qquad \mu_{ES}^{3.2}(u,0,5,9) = \begin{cases} 0, \, u < 0; \\ \frac{u}{5}, 0 \le u \le 5; \\ \frac{9-u}{4}, 5 \le u \le 9; \\ 0, u \ge 9; \end{cases}$$

$$\mu_{ES}^{3.3}(u,0,2,9) = \begin{cases} 0, \, u < 0; \\ \frac{u}{2}, 0 \le u \le 2; \\ \frac{9-u}{7}, 2 \le u \le 9; \\ 0, u \ge 9; \end{cases}$$

The graphs of these functions, shown in the MATLAB system environment of the Fuzzy Logic Toolbox package, have the form shown in Figure 4.

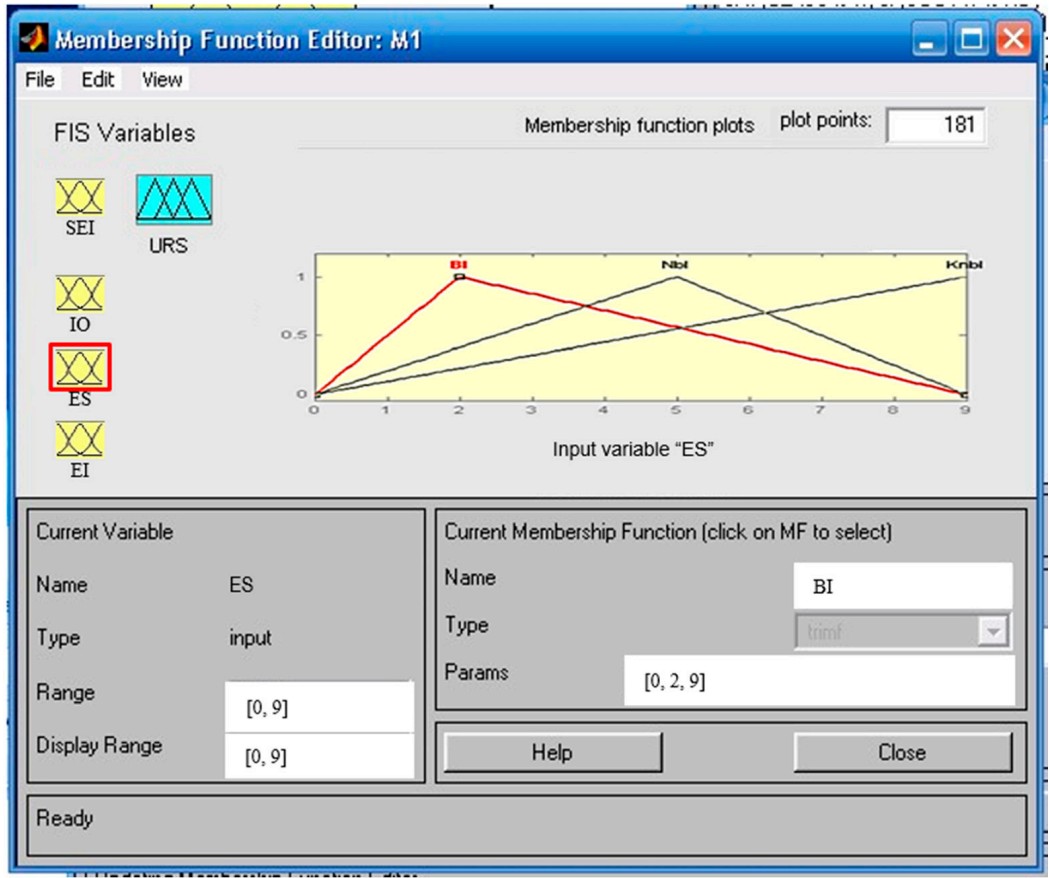

**Figure 4.** Graphs of membership functions of the fuzzy sets of a linguistic variable *ES*.

Semantics of fuzzy sets $A_{EI}^i \in T(EI)$ in the form of membership functions $\mu_{EI} = \{\mu_{A_{EI}^i}\}_{i=1}^{\alpha}$ of trapezoidal type, assuming that their universums are segments: $U_{EI} = [0, 9]$: $\mu_{EI}^{4.1}(u, 0, 9, 9)$, $\mu_{EI}^{4.2}(u, 0, 6, 9)$, $\mu_{EI}^{4.3}(u, 0, 3, 9)$, $\mu_{EI}^{4.4}(u, 0, 1, 9)$. The membership functions entered have the following form:

$$\mu_{EI}^{4.1}(u, 0, 9, 9) = \begin{cases} 0, u < 0; \\ \frac{u}{9}, 0 \le u \le 9; \\ 0, u > 9; \end{cases} \qquad \mu_{EI}^{4.2}(u, 0, 6, 9) = \begin{cases} 0, u < 0; \\ \frac{u}{6}, 0 \le u \le 6; \\ \frac{9-u}{3}, 6 \le u \le 9; \\ 0, u \ge 9; \end{cases}$$

$$\mu_{EI}^{4.3}(u, 0, 3, 9) = \begin{cases} 0, u < 0; \\ \frac{u}{3}, 0 \le u \le 3; \\ \frac{9-u}{6}, 3 \le u \le 9; \\ 0, u \ge 9; \end{cases} \qquad \mu_{EI}^{4.4}(u, 0, 1, 9) = \begin{cases} 0, u < 0; \\ \frac{u}{1}, 0 \le u \le 1; \\ \frac{9-u}{8}, 1 \le u \le 9; \\ 0, u \ge 9. \end{cases}$$

In the MATLAB system of the Fuzzy Logic Toolbox package, these functions take the form shown in Figure 5.

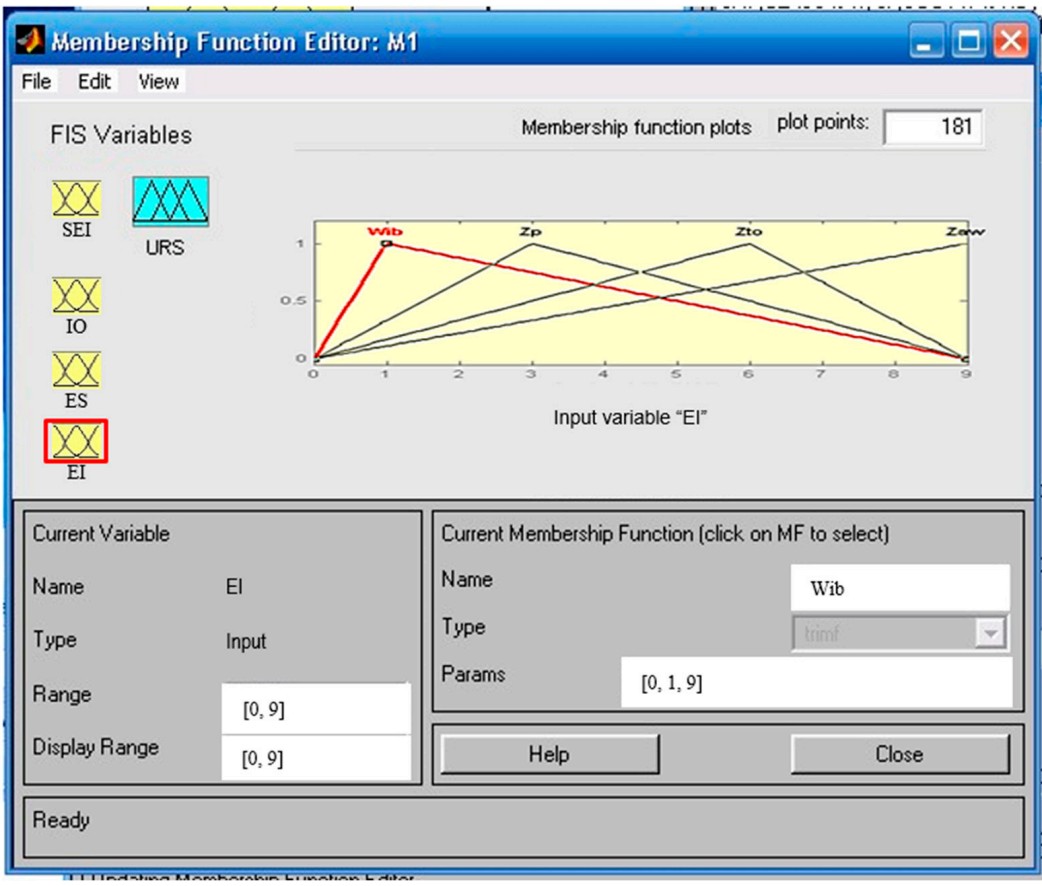

**Figure 5.** Graphs of membership functions of fuzzy sets of a linguistic variable *EI*.

Analytical expressions for the membership functions of fuzzy sets included in the linguistic variable *URS* are:

$$\mu_{URS}^{L}(u, 0, 1, 3) = \begin{cases} 0, \, u < 0; \\ \frac{u}{1}, 0 \le u \le 1; \\ \frac{3-u}{2}, 1 \le u \le 3; \\ 0, u \ge 3; \end{cases} \qquad \mu_{URS}^{M}(u, 0, 2, 3) = \begin{cases} 0, \, u < 0; \\ \frac{u}{2}, 0 \le u \le 2; \\ \frac{3-u}{1}, 2 \le u \le 3; \\ 0, u \ge 3; \end{cases}$$

$$\mu_{URS}^{H}(u, 0, 3, 3) = \begin{cases} 0, \, u < 0; \\ \frac{u}{3}, 0 \le u \le 3; \\ 0, u > 3; \end{cases}$$

The graphs of these functions, shown in the MATLAB system environment of the Fuzzy Logic Toolbox package, have the form shown in Figure 6.

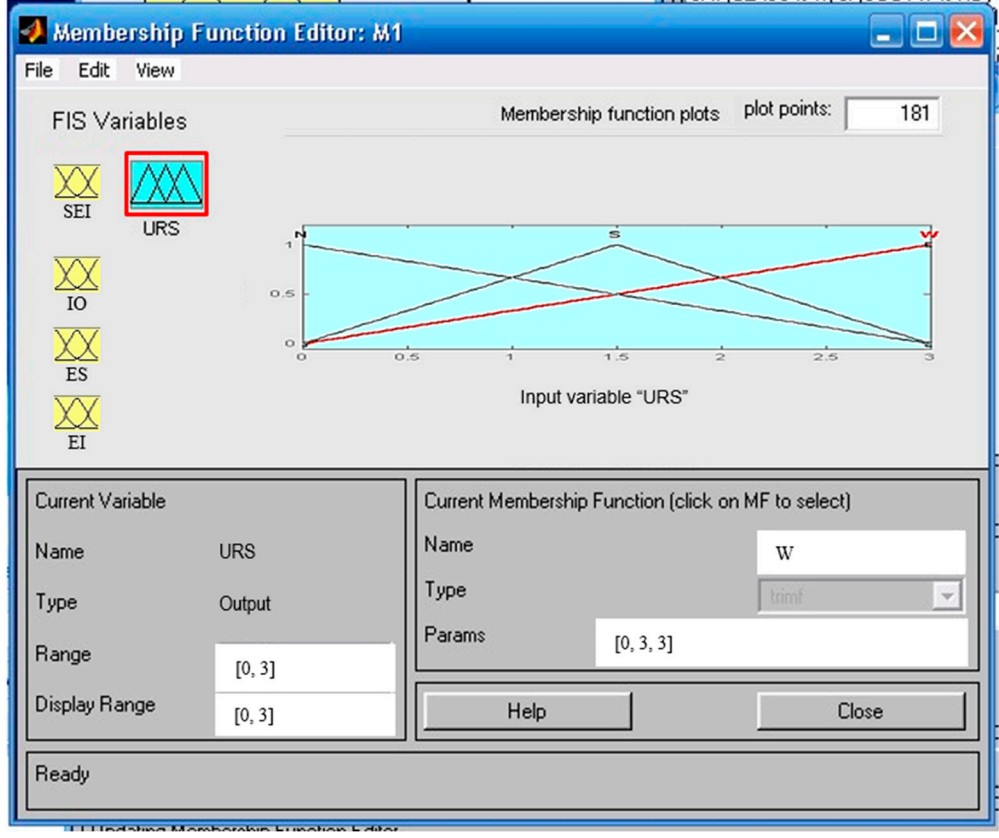

**Figure 6.** Graphs of membership functions of fuzzy sets of a linguistic variable *URS*.

The verbal form of the inference rules system used in the model, as formed based on knowledge and experience of experts, have the following form:

- If *SEI* is 1.1 and *IO* is 2.1 and *ES* is 3.1 then *URS* is *H*;
- If *SEI* is 1.2 and *IO* is 2.1 and *ES* is 3.1 then *URS* is *H*;
- If *SEI* is 1.3 and *IO* is 2.1 and *ES* is 3.1 then *URS* is *H*;
- If *SEI* is 1.4 and *IO* is 2.1 and *ES* is 3.1 then *URS* is *H*;
- If *ES* is 3.1 and *EI* is 4.2 then *URS* is *H*;
- If *ES* is 3.2 and *EI* is 4.2 then *URS* is *H*;
- If *SEI* is 1.4 and *ES* is 3.3 and *EI* is 4.4 then *URS* is *M*;
- If *ES* is 3.3 and *EI* is *not* (4.1) then *URS* is L;
- If *ES* is 3.3 and *EI* is *not* (4.2) then *URS* is L;
- If *ES* is 3.3 and *EI* is *not* (4.3) then *URS* is L;
- If *ES* is 3.3 and *EI* is *not* (4.4) then *URS* is L.

Figure 7 shows the graphical form of the model for assessing the project's compliance with environmental criteria.

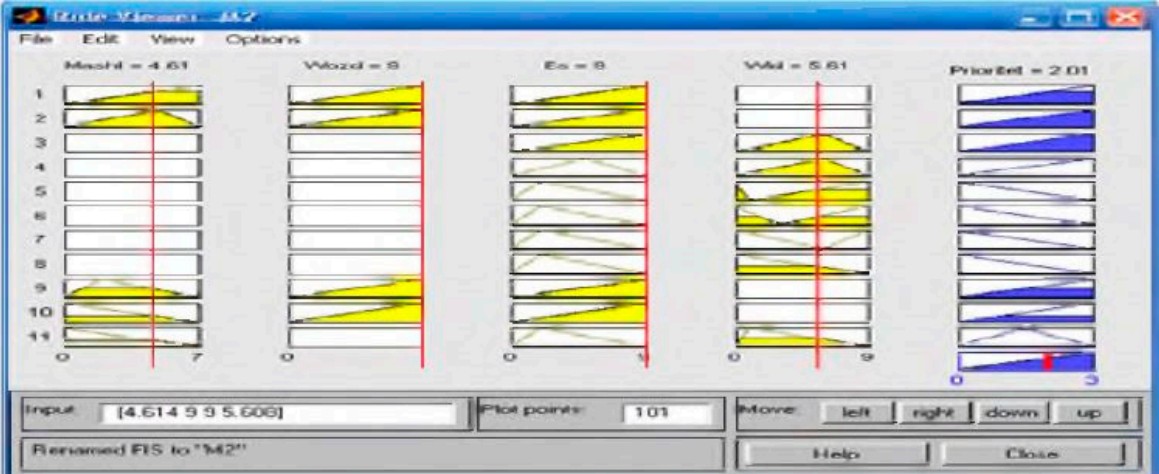

**Figure 7.** The result of the operation of the model for assessing the project's compliance with environmental criteria.

By setting any values of the indicators in the input to the model $\{SEI, IO, ES, EI\}$, the expert can easily obtain an integrated assessment of the investment project submitted for financing. Figure 7 shows that for the integral index of $SEI = 4.61$, $IO = 9$, $ES = 9$, $EI = 5.61$, the indicator of the level of compliance with environmental criteria $URS = 2.01$ is established. The model for assessing the project's compliance with environmental criteria is able to adapt to changes in both the rules of logical inference and the membership functions of the local characteristics of the project, as based on the knowledge of experts.

## 5. Conclusions

The article presents the following results, which have scientific novelty:

(a) A new approach to the design and environmental analysis of investment projects at the stage of environmental screening, as distinct from the existing inclusion of economic and mathematical models in the chain of analysis of processing semi-structured results of expert assessments. The advantage of the approach is the ability to formally describe and use expert knowledge to evaluate investment projects in terms of the imperatives of economic and environmental development of the region.

(b) A complex of economic and mathematical models of design and investment analysis at the stage of environmental screening, as distinct from the existing application of the mathematical apparatus of fuzzy algebra and fuzzy logic. The advantage of the models is that one has the possibility of quantitative processing of qualitative information, as reflecting the semi-structured knowledge of specialists.

**Author Contributions:** All authors contributed equally to this paper.

**Funding:** This research received no external funding.

**Conflicts of Interest:** The authors declare no conflict of interest.

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
