# Peer review of "Assessment of Investment Attractiveness of Projects on the Basis of Environmental Factors"

_sustainability, doi:10.3390/su11092544_

Round 1
Reviewer 1 Report
Authors describe the use of fuzzy logic in project's evaluation process in the scope of environmental sustainability . My major comments are listed below:
- the description of PROBLEM SOLVING METHOD (see section 3) is quite unreadable (see p.5,6,7);
- section 4 should be extended with a study case of a project evaluation;
- the beginning of the paper should be extended with current approaches used for project's evaluation;
- advantages of the proposed method should be exposed compared to current approaches;
- the title of the paper should be narrowed to the described method;
Author Response
- the description of PROBLEM SOLVING METHOD (see section 3) is quite unreadable (see p.5,6,7);
Corrected, in the text highlighted with yellow, all text re—edited by Dr. Mark Watkins, mjw64@leicester.ac.uk
- section 4 should be extended with a study case of a project evaluation;
Corrected, in the text highlighted with yellow.
- the beginning of the paper should be extended with current approaches used for project's evaluation;
Corrected.
- advantages of the proposed method should be exposed compared to current approaches;
corrected
- the title of the paper should be narrowed to the described method;
Corrected. New title «ASSESSMENT OF INVESTMENT ATTRACTIVENESS OF PROJECTS ON THE BASIS OF ENVIRONMENTAL FACTORS»

Reviewer 2 Report
There are many grammatical errors that need to be fixed in the paper All in-line formulas / symbols need to have the same font size as other texts Equation (1) is very confusing, will need more explanations for the readers to understand How were all the MFs determined in Section 4? What was the inference engine used in this example? Why did you choose this one? Also, I don’t quite get how the model (CLASS * SOOTW →URS) works. At the beginning, CLASS is defined as {A, B, W, G}, which are linguistic terms; however, in the example, CLASS = 0.596 was used. What does it mean for CLASS = 0.596? The same as other variables. The knowledge base (rule base) defined in this paper is already comprehensive, meaning for every single combination of class and sootw, there is a corresponding URS as the output. I am not sure why fuzzy sets are needed in this problem, nor a fuzzy inference model.
Author Response
We thank the Reviewer for her/his interest in our work and for helpful comments that will greatly improve the manuscript and we have tried to do our best to respond to the points raised. The Reviewer has brought up some good points and we appreciate the opportunity to clarify our research objectives and results. As indicated below, we have checked all the general and specific comments provided by the Reviewer and have made necessary changes accordingly to her/his indications. We will detail in our response below.
Comments:
There are many grammatical errors that need to be fixed in the paper.
Response: We appreciate your comments. We have checked Comments provided by the Reviewer and have made necessary changes accordingly to her/his indications. All text re—edited by Dr. Mark Watkins, mjw64@leicester.ac.uk

Reviewer 3 Report
A really nice manuscript on a rather novel topic area. It would have been nice to have at least one worked example for either a real or fictitious investment project.
Furthermore, the sections from line 315 to 325 and 329 to 335 need to be completed correctly and appropriately.
Author Response
We thank the Reviewer for her/his interest in our work and for helpful comments that will greatly improve the manuscript and we have tried to do our best to respond to the points raised. The Reviewer has brought up some good points and we appreciate the opportunity to clarify our research objectives and results. As indicated below, we have checked all the general and specific comments provided by the Reviewer and have made necessary changes accordingly to her/his indications. We will detail in our response below.
A really nice manuscript on a rather novel topic area. It would have been nice to have at least one worked example for either a real or fictitious investment project.
Response: We appreciate your comments.

Reviewer 4 Report
string 185 - UCLASS = [1,10]
string 185 - USOOTW = [0,5]
Need to explain why this value - 10, 5 ? What does this mean?
Links to articles about fuzzy expert systems are needed, in which similar tasks the same apparatus is used?
Author Response
We thank the Reviewer for her/his interest in our work and for helpful comments that will greatly improve the manuscript and we have tried to do our best to respond to the points raised. The Reviewer has brought up some good points and we appreciate the opportunity to clarify our research objectives and results. As indicated below, we have checked all the general and specific comments provided by the Reviewer and have made necessary changes accordingly to her/his indications. We will detail in our response below.

Round 2
Reviewer 1 Report
All my suggestions (remarks) had been addressed adequately.
Author Response
We thank the Reviewer for her/his interest in our work and for helpful comments that will greatly improve the manuscript and we have tried to do our best to respond to the points raised. The Reviewer has brought up some good points and we appreciate the opportunity to clarify our research objectives and results. As indicated below, we have checked all the general and specific comments provided by the Reviewer and have made necessary changes accordingly to her/his indications. We will detail in our response below.
Comments:
All my suggestions (remarks) had been addressed adequately.
Response: We appreciate your comments. Thank you

Reviewer 2 Report
The authors addressed a lot of my comments from last review and added much more details of the problem formulations and symbols used in the paper, now it is much more understandable.
There are equation (1) in the paper. Also, they are not technically "equations", you might consider state them as "Definitions".
The explanation after (1) on page 10 is still a bit confusing: there is a symbol alpha which was never described; and, symbol A's were never explained neither.
The Matlab screenshots are very vague, I can not see the texts in the figures at all.
I still have troubles accepting the choices of all the MFs. They all seem very similar to me, especially adjacent ones. For example, for the two MFs (I really can't see what are they representing from the figure) in the middle in Fig. 3, I highly doubt using only one of them in the system and removing the corresponding rules will make any difference compared to keep both, since the MFs are almost the same.
Following up with my last comment, can you show more simulation results with different input combinations?
Author Response
We thank the Reviewer for her/his interest in our work and for helpful comments that will greatly improve the manuscript and we have tried to do our best to respond to the points raised. The Reviewer has brought up some good points and we appreciate the opportunity to clarify our research objectives and results. As indicated below, we have checked all the general and specific comments provided by the Reviewer and have made necessary changes accordingly to her/his indications. We will detail in our response below.
Comments:
There are equation (1) in the paper. Also, they are not technically "equations", you might consider state them as "Definitions".
Response: We appreciate your comments. Definitions (1) is not an equation, but a tuple whose elements are the economic-mathematical models developed in the article.

Reviewer 3 Report
Appropriate changes made in light of reviewer's comments
Round 3
Reviewer 2 Report
The authors addressed all my comments and I think the paper is now in a good state.